# Pd-Catalyzed Hirao P–C Coupling Reactions with Dihalogenobenzenes without the Usual P-Ligands under MW Conditions

Bianka Huszár [1] , Petra Regina Varga [1], Nóra Á. Szűcs [1], András Simon [2], László Drahos [3] and György Keglevich [1,*]

1 Department of Organic Chemistry and Technology, Budapest University of Technology and Economics, 1521 Budapest, Hungary
2 Department of Inorganic and Analytical Chemistry, Budapest University of Technology and Economics, 1521 Budapest, Hungary
3 MS Proteomics Research Group, Research Centre for Natural Sciences, 1117 Budapest, Hungary
* Correspondence: keglevich.gyorgy@vbk.bme.hu; Tel.: +36-1-463-1111 (ext. 5883)

**Abstract:** A literature survey of the P–C coupling reactions of 1,4-and 1,2-bromo-iodobenzenes with diphenylphosphine oxide or diethyl phosphite under different conditions comprising Pd-, Ni-, or Cu-catalysis revealed that, depending on the experimental details, the yields of the corresponding >P(O)-bromobenzenes were rather diverse and occasionally contradicting. Therefore, the reactivity of a series of 1,4-, 1,3- and 1,2-dibromo- and bromo-iodobenzenes with the above mentioned P-reagents was evaluated under the "P-ligand-free" microwave (MW)-assisted conditions elaborated by us. Starting from dibromobenzenes and iodo-bromoarenes, practical and competent syntheses were developed for phosphonoyl- and phosphinoyl-bromoarenes, and, in a few instances, for arenes with two P-functions. The cheaper dibromobenzenes may be substituted for the bromo-iodo derivatives. In all, 12 products were prepared in yields of 45–82%. They were fully characterized. The method described does not require the use of traditional P-ligands.

**Keywords:** Hirao reaction; P–C coupling; dihalogenobenzenes; Pd; selectivity





## 1. Introduction

For the preparation of dialkyl arylphosphonates, the Hirao reaction comprising the P–C coupling between a dialkyl phosphite and an aryl or vinyl halide (mostly bromide) using palladium-tetrakistriphenylphosphine as the catalyst and triethylamine as the base in different solvents is a convenient method [1–8]. The original procedure was followed using variations applying in situ formed Pd catalysts from suitable precursors (e.g. Pd(OAc)$_2$ and PdCl$_2$) and added mono- or bidentate P-ligands. The reaction model was extended to *H*-phosphinates and secondary phosphine oxides using different arenes [9–15]. Keglevich et al. developed a microwave-assisted method in which the excess of the >P(O)H reagent served as the P-ligand, via its trivalent tautomeric form, to Pd [16–20]. Hence, there was no need to apply traditional P-ligands decreasing both the cost and the environmental burden.

The data accumulated on the P–C coupling reaction of bromo-iodobenzenes are summarized in Table 1. 1-Bromo-4-iodobenzene was reacted with diphenylphosphine oxide and diethyl phosphite, applying palladium-tetrakistriphenylphosphine as the catalyst. In the two cases, the base, solvent and the temperature were NEt$_3$/PhMe/110 °C and Cs$_2$CO$_3$/THF/120 °C, respectively (Table 1, entries 1 and 2). The latter experiment was performed on MW irradiation. The 4-bromophenylphosphine oxide (**I**, Y = Ph) and 4-bromophenylphosphonate (**I**, Y = EtO) were obtained in a yield of 85% and 19%, respectively [21,22]. The latter product could be better prepared from the reaction of 4-bromo-iodobenzene with (EtO)$_2$P(O)H using Pd(OAc)$_2$/dppf catalyst with NEt$_3$ in THF at 68 °C

in the presence of KOAc as the additive. The phosphonate (**I**, Y = EtO) was obtained in a somewhat higher yield of 48% (Table 1, entry 3) [15,23].

1-Bromo-2-iodobenzene was also used in Hirao P–C coupling. The reaction with Ph$_2$P(O)H was performed using Pd(dba)$_2$/dppp and Pd$_2$(dba)$_3$/dppp and DIPEA in PhMe at 120 °C or 90 °C to afford the corresponding 2-bromophenylphosphine oxide (**I**, Y = Ph) in yields of 65 and 64%, respectively (Table 1, entries 4 and 5) [24,25], while the coupling with (EtO)$_2$P(O)H was realized applying Pd(OAc)$_2$/PPh$_3$ and DIPEA in EtOH at reflux. The 2-bromophenylphosphonate (**I**, Y = EtO) was obtained in 74% yield (Table 1, entry 6) [26]. The relatively good yields are somewhat surprising, if the steric hindrance in the initial dihalogenobenzene is considered. Although we applied the same reaction conditions (Pd(OAc)$_2$/PPh$_3$, DIPEA, EtOH, reflux) [26], the results of the experiment shown under entry 6 of Table 1 could not be reproduced. Practically, no 2-bromo-diethylphosphonoylbenzene was formed.

The above dihalogenobenzene → P(O)H reactions were also carried out utilizing Ni- and Cu-catalysts. Regarding the application of the NiCl$_2$/2,2'-bipyridyl catalyst in the P–C coupling of 1,4-dibromobenzene or 4-bromo-iodobenzene and Ph$_2$P(O)H, Zn was also added in the reaction mixture [27]. In these cases, the bisphosphinoyl species was the target. However, we proved in another study, that the addition of a metal reducing agent into a Ni-catalyzed Hirao reaction causes problems instead of promoting the P–C coupling [28]. Consequently, the 75–88% yields reported may be questioned [27]. In another variation, 4-bromo-iodobenzene was reacted with Ph$_2$P(O)H using Ni(cod)$_2$/dtbbpy along with Ru(bpy)$_3$Cl$_2$ as the co-catalyst and Cs$_2$CO$_3$ as the base in MeOH at 23 °C on blue-LED irradiation to give the corresponding diphenyl-arylphosphine oxide in an 81% yield (Table 1, entry 7) [29].

The coupling of 1-bromo-2-halogenobenzenes with Ph$_2$P(O)H using CuI/$\alpha$-phenylethylamine in the presence of K$_2$CO$_3$ in toluene at 110 °C was also investigated [30]. Under such conditions, starting from 1,2-dibromobenzene, no formation of the expected bis(phosphinoyl) product was observed. However, applying 1-bromo-2-iodobenzene along with one equivalent of Ph$_2$P(O)H, the bis(phosphinoyl) product was isolated in a yield of 43% [30]. A similar reaction of 1-bromo-2-iodobenzene with (EtO)$_2$P(O)H in the presence of CuI applied along with different N-ligands and Cs$_2$CO$_3$ in toluene at 110 °C afforded the 4-bromophenylphosphonate (**I**, Y = EtO) in up to an 85% yield (Table 1, entries 8 and 9) [31,32].

It may be seen that applying the classical catalyst Pd(PPh$_3$)$_4$, 1-bromo-4-iodobenzene could be efficiently converted to the 4-bromophenyl-diphenylphosphine oxide. However, the yield of the 4-bromophenylphosphonate was low, even when using Pd(OAc)$_2$/dppf as the catalyst. The latter species was obtained in better yields applying a Cu catalyst with N-ligands. Surprisingly, the 2-bromophenyl-phosphine oxide and phosphonate were isolated in rather good (64–74%) yields using different Pd/phosphine catalysts starting with 1-bromo-2-iodobenzene. In any case, the outcome of the P–C coupled products was variable. We wished to evaluate how the different bromo-halogenobenzenes including dibromo derivatives may be converted to phosphine oxides and phosphonates applying the MW-assisted "P-ligand-free" approach developed previously by us [17,18]. On the other hand, our purpose was to synthesize bromophenyl-phosphine oxides and phosphonates, suitable starting materials in Arbuzov reactions and Suzuki cross couplings.

**Table 1.** P–C coupling reactions of bromo-iodobenzenes in the presence of different catalysts/precursors – a literature survey.

| Entry | ArX$^1$X$^2$ | P-Reagent | Catalyst (Precursor)/Ligand | Additional Conditions | Base | Solvent | T (°C) | Isolated Yield (%) | Ref. |
|---|---|---|---|---|---|---|---|---|---|
| 1 | 1-Br-4-IC$_6$H$_4$ | Ph$_2$P(O)H | Pd(PPh$_3$)$_4$ | – | NEt$_3$ | PhMe | 110 | 85 | [21] |
| 2 | 1-Br-4-IC$_6$H$_4$ | (EtO)$_2$P(O)H | Pd(PPh$_3$)$_4$ | MW | Cs$_2$CO$_3$ | THF | 120 | 19 | [22] |
| 3 | 1-Br-4-IC$_6$H$_4$ | (EtO)$_2$P(O)H | Pd(OAc)$_2$/dppf | KOAc additive | NEt$_3$ | THF | 68 | 48 | [15,23] |
| 4 | 1-Br-2-IC$_6$H$_4$ | Ph$_2$P(O)H | Pd(dba)$_2$/dppp | – | DIPEA | PhMe | 120 | 65 | [24] |
| 5 | 1-Br-2-IC$_6$H$_4$ | Ph$_2$P(O)H | Pd$_2$(dba)$_3$/dppp | – | DIPEA | PhMe | 90 | 64 | [25] |
| 6 | 1-Br-2-IC$_6$H$_4$ | (EtO)$_2$P(O)H | Pd(OAc)$_2$/PPh$_3$ | – | DIPEA | EtOH | reflux | 74 | [26] |
| 7 | 1-Br-4-IC$_6$H$_4$, | Ph$_2$P(O)H | Ni(cod)$_2$/dtbbpy, Ru(bpy)$_3$Cl$_2$·6H$_2$O | blue LED | Cs$_2$CO$_3$ | MeOH | 26 | 81 | [29] |
| 8 | 1-Br-4-IC$_6$H$_4$ | (EtO)$_2$P(O)H | CuI/phen | – | Cs$_2$CO$_3$ | PhMe | 100 | 84 | [31] |
| 9 | 1-Br-4-IC$_6$H$_4$ | (EtO)$_2$P(O)H | CuI/proline or pipecolonic acid | – | Cs$_2$CO$_3$ | PhMe | 110 | 86/85 | [32] |

## 2. Results and Discussion

First, the reaction of 1,4-dibromobenzene with diphenylphosphine oxide was investigated in the presence of 5 mol% Pd(OAc)$_2$, and 1.1 equivalents of triethylamine in ethanol as the solvent under on microwave irradiation. According to our earlier protocol, the Ph$_2$P(O)H was measured in a 1.15 equivalents quantity to ensure the P-ligand and the reducing agent for the Pd(0)→Pd(II) transition [18]. After an irridation at 120 °C for 30 min, the mixture contained 85% of the expected 4-bromo(phenyl-diphenylphosphine oxide **1a**, 6% of the bisphosphinoyl compound **2a** along with 7% of (EtO)Ph$_2$P(O) and 2% of Ph$_3$P(O) by-products (Table 2, entry 1). Preparative yield of the target product was 63% after chromatography. Repeating the reaction applying 2.15 equivalents of the P-reagent and 1 h reaction time, the mixture comprised 15% of monophosphinoylarene **1a**, 41% of bisphosphinoylbenzene **2a**, 16% of (EtO)Ph$_2$P(O) and 26% of Ph$_3$P(O) beside 2% of the unreacted starting material (Ph$_2$P(O)H) (Table 2, entry 2). One can see that the latter reaction was not selective. Changing for diethyl phosphite, after an irradiation at 120 °C for 1 h, the diethylphosphonoylarene **1b** was present in 69% together with 18% of bis(diethylphosphono)benzene **2b** and 12% of (EtO)$_2$PhP(O) (Table 2, entry 3). The desired product **1b** was obtained in a yield of 48% after chromatography.

**Table 2.** P–C coupling reactions of 1,4-dibromobenzene.

| Entry | Y | P-Reagent (equiv.) | t (min) | Conversion (%) [a] | Product Composition (%) [a] | | | | Isolated Yield of 1 (%) |
|---|---|---|---|---|---|---|---|---|---|
| | | | | | 1 | 2 | Ph$_2$(EtO)P(O) (A) or Ph(EtO)$_2$P(O) (B) | Ph$_3$P(O) | |
| 1 | Ph (**a**) | 1.15 | 30 | 100 | 85 | 6 | 7 (A) | 2 | 63 (**1a**) |
| 2 | Ph (**a**) | 2.15 | 60 | 98 | 15 | 41 [b] | 16 (A) [c] | 26 [d] | – |
| 3 | EtO (**b**) | 1.15 | 60 | 99 | 69 | 18 [e] | 12 (B) [f] | n. r. | 48 (**1b**) |

[a] Based on $^{31}$P NMR. [b] $\delta_P$ (CDCl$_3$) 29.8, $\delta_P$ lit. [27] (CDCl$_3$) 36.8; [M + H]$^+$ = 479.1323 C$_{30}$H$_{25}$O$_2$P$_2$ requires 479.1330. [c] $\delta_P$ (CDCl$_3$) 31.4, $\delta_P$ lit. [17] (CDCl$_3$) 32.2; [M + H]$^+$ = 247.0882, C$_{14}$H$_{16}$O$_2$P requires 247.0888. [d] $\delta_P$ (CDCl$_3$) 29.2, $\delta_P$ lit. [28] (CDCl$_3$) 29.5; [M + H]$^+$ = 279.0941, C$_{18}$H$_{16}$OP requires 279.0939. [e] $\delta_P$ (CDCl$_3$) 16.9, $\delta_P$ lit. [33] (CDCl$_3$) 16.8; [M + H]$^+$ = 351.1121 C$_{14}$H$_{25}$O$_6$P$_2$ requires 351.1126. [f] $\delta_P$ (CDCl$_3$) 18.8, $\delta_P$ lit. [32] (CDCl$_3$) 19.4; [M + H]$^+$ = 215.0827, C$_{10}$H$_{16}$O$_3$P requires 215.0837. n. r.: not relevant.

The experiments with the 4-bromo-iodobenzene were performed in a similar manner; however, the formation of 4-bromophenyl-diphenylphosphine oxide (**1a**) and diethylphosphono-bromobenzene (**1b**) was more selective at 120 °C as in the previous cases (see Table 2, entries 1 and 3) indicated by their proportions of 97% and 88%, respectively (Table 3, entries 2 and 4). The coupling with Ph$_2$P(O)H led to similar results after an irradiation at 100 °C for 1 h (Table 3, entry 1). Using 2.15 equivalents of the secondary phosphine oxide, the selectivity was similar to the reaction with 1,4-dibromobenzene (19% of **1a**, and 42% of **2a** were formed) (Table 3, entry 3). At the same time, in reaction with two equivalents of (EtO)$_2$P(O)H, the coupling was significantly more selective, as indicated by the 87:13 ratio of products **2b** and **1b**. It is noteworthy that the latter conversion took place without any side reaction (Table 3, entry 5).

**Table 3.** P–C coupling reactions of 4-bromo-iodobenzene.

| Entry | Y | P-Reagent (equiv.) | T (°C) | t (min) | Conversion (%) [a] | Product Composition (%) [a] | | | | Isolated Yield of 1 (%) |
|---|---|---|---|---|---|---|---|---|---|---|
| | | | | | | 1 | 2 | Ph$_2$(EtO)P(O) (A) or Ph(EtO)$_2$P(O) (B) | Ph$_3$P(O) | |
| 1 | Ph (a) | 1.15 [b] | 100 | 60 | 100 | 96 | – | 2 (A) | 2 | 74 (**1a**) |
| 2 | Ph (a) | 1.15 [b] | 120 | 30 | 100 | 97 | – | 2 (A) | 1 | 75 (**1a**) |
| 3 | Ph (a) | 2.15 [b] | 120 | 60 | 100 | 19 | 42 | 8 (A) | 31 | – |
| 4 | EtO (b) | 1.15 [b] | 120 | 30 | 98 | 88 | 7 | 3 (B) | n. r. | 63 (**1b**) |
| 5 | EtO (b) | 2.3 [c] | 120 | 30 | 100 | 13 | 87 | – | n. r. | 65 (**2b**) |

[a] Based on $^{31}$P NMR. [b] In the presence of 5 mol% Pd(OAc)$_2$ catalyst. [c] In the presence of 10 mol% Pd(OAc)$_2$ catalyst. n. r.: not relevant.

The application of 1,4-dibromobenzene as the starting material provided products **1a** and **1b** in lower yields as compared to the cases starting from 4-bromo-1-iodobenezene. Our method afforded aryldiphenylphosphine oxide (**1a**) in a somewhat lower yield (63–75%) than that obtained using the literature method (85%) [21]. However, the phosphonoarene **1b** was obtained in a higher yield (63%) as compared to the outcomes (19 and 48%) reported [15,22,23].

Changing for 1,3-dibromobenzene and using $Ph_2P(O)H$ under the conditions applied above, 3-bromophenyl-diphenylphosphine oxide **3a** was formed in a similar selectivity as the para analogue **1a** (Table 4, entry 1). At the same time, applying 2.15 equivalents of the P-reagent, the bisphosphinoylbenzene (**4a**) was formed in a selectivity of 92% (Table 4, entry 2). Products **3a** and **4a** were isolated in a 68% and 75% yield, respectively, after chromatography. The reaction of the dibromobenzene and 1.15 equivalents of $Ph_2P(O)H$ (Table 4, entry 1) was repeated on a 3-fold scale. In this case, the yield of compound **3a** increased to 76%. The coupling of 1,3-dibromobenzene with 1.15 equivalents of diethyl phosphite was not especially selective, either at 120 °C/1 h in EtOH, or at 150 °C/30 min (solvent-free), as the ratio of species **3b** and **4b** was 57:27 and 63:24, respectively (Table 4, entries 3 and 4). Diethylphosphono-bromobenzene (**3b**) was obtained in a 45% yield after purification. The P–C coupling with $Ph_2P(O)H$ seemed to be easier and more selective than that with $(EtO)_2P(O)H$.

**Table 4.** P–C coupling reactions of 1,3-dibromobenzene.

| Entry | Y | P-Reagent (equiv.) | T (°C) | t (min) | Solvent | Conversion (%) [a] | Product Composition (%) [a] | | | | Isolated Yield (%) |
|---|---|---|---|---|---|---|---|---|---|---|---|
| | | | | | | | 3 | 4 | $Ph_2(EtO)P(O)$ (A) or $Ph(EtO)_2P(O)$ (B) | $Ph_3P(O)$ | |
| 1 | Ph (**a**) | 1.15 | 120 | 25 | EtOH | 100 | 85 | 6 | 4 (**A**) | 5 | 68 (**3a**) |
| 2 | Ph (**a**) | 2.15 | 120 | 30 | EtOH | 100 | 2 | 92 | 2 (**A**) | 4 | 75 (**4a**) |
| 3 | EtO (**b**) | 1.15 | 120 | 60 | EtOH | 94 | 57 | 27 [b] | 10 (**B**) | n. r. | – |
| 4 | EtO (**b**) | 1.15 | 150 | 30 | – | 97 | 63 | 24 [b] | 10 (**B**) | n. r. | 45 (**3b**) |

[a] Based on $^{31}P$ NMR. [b] $\delta_P$ (CDCl$_3$) 16.9; $[M + H]^+ = 351.1123$ $C_{14}H_{25}O_6P_2$ requires 351.1126. n. r.: not relevant.

Repeating the reactions with 3-bromo-iodobenzene, all reactions were more selective, if not completely selective. The bromophenyl-diphenylphosphine oxide **3a** was present in the reaction mixture in 94–95%, regardless of whether 100 °C/1 h or 120 °C/30 min was applied (Table 5, entries 1 and 2). Product **3a** was isolated in 78 and 75% yields. Measuring in 2.15 equivalents of $Ph_2P(O)H$, after irradiation at 120 °C for 35 min, the proportion of bisphosphinoylbenzene **4a** was 88%, which could be isolated in a yield of 70% (Table 5, entry 3). The diethylphosphono-bromobenzene **3b** synthesized at 120 °C/30 min was present in 81% in the crude mixture, which could be prepared in a 66% yield (Table 5, entry 4).

**Table 5.** P–C coupling reactions of 3-bromo-iodobenzene.

| Entry | Y | P-Reagent (equiv.) | T (°C) | t (min) | Solvent | Conversion (%) [a] | Product Composition (%) [a] | | | | Isolated Yield (%) |
|---|---|---|---|---|---|---|---|---|---|---|---|
| | | | | | | | 3 | 4 | Ph$_2$(EtO)P(O) (A) or Ph(EtO)$_2$P(O) (B) | Ph$_3$P(O) | |
| 1 | Ph (**a**) | 1.15 | 100 | 60 | EtOH | 100 | 94 | – | 1 (A) | 5 | 78 (**3a**) |
| 2 | Ph (**a**) | 1.15 | 120 | 30 | EtOH | 100 | 95 | 3 | 1 (A) | 2 | 75 (**3a**) |
| 3 | Ph (**a**) | 2.15 | 120 | 35 | EtOH | 97 | – | 88 | 3 (A) | 6 | 70 (**4a**) |
| 4 | EtO (**b**) | 1.15 | 120 | 30 | – | 98 | 81 | 13 | 4 (B) | n. r. | 66 (**3b**) |

[a] Based on $^{31}$P NMR. n. r.: not relevant.

The MW-assisted "P-ligand-free" Hirao reaction is a good choice for the preparation of 3-bromophenyl-P(O)Y$_2$ products **3a** and **3b**. 1,3-Dibromobenzene is also a suitable starting material, however, the 1-iodo-3-bromobenzene for more efficient preparations. The latter could also be converted to the bis-phosphinoyl derivative (**4a**).

3-Chloro-bromobenzene could also be applied to prepare the corresponding 1-phosphinoyl- and 1-phosphonoyl-3-chlorobenzenes **5a** and **5b**, respectively, in a selective manner (Table 6, entries 1 and 2). However, longer reaction times were necessary at 120 °C as compared to the cases, in which 3-bromo-1-iodobenzene was used as the starting material.

**Table 6.** P–C coupling reactions of 3-chloro-bromobenzene.

| Entry | Y | t (min) | Isolated Yield of 5 (%) |
|---|---|---|---|
| 1 | Ph (**a**) | 60 | 80 (**5a**) |
| 2 | EtO (**b**) | 45 | 82 (**5b**) |

The coupling of 1,2-dibromobenzene with 1.15 equivalents of diphenylphosphine oxide in the presence of 5 mol% of Pd(OAc)$_2$ and 1.1 equivalents triethylamine in ethanol as the solvent at 150 °C on MW irradiation was rather inefficient. As can be seen from Table 7, entry 1, the 2-bromo-phosphinoylbenzene (**6a**) was formed in 20%, while the di(Ph$_2$P(O))-product (**7a**) was also present in a comparable proportion (19%). The remaining part covered 48% of triphenylphosphine oxide (48%) formed on debromination, and phenylphosphinic acid coming from the oxidation of Ph$_2$P(O)H. An increase in the quantity of the catalyst precursor from 5 to 10 mol% and that of the P-species from 1.15 to 1.3 favored the formation of Ph$_3$P(O) (58%), otherwise there was no significant change except that some ethyl diphenylphosphinate also appeared in the mixture (Table 7, entry 2).

**Table 7.** P–C coupling reactions of 1,2-dibromobenzene.

| Entry | Y | P-Reagent (equiv.) | t (min) | Solvent | Conversion (%) [a] | Product Composition (%) [a] | | | | Isolated Yield of 6 (%) |
|-------|---|--------------------|---------|---------|--------------------|---|---|---|---|-------------------------|
| | | | | | | **6** | **7** | Ph$_2$(EtO)P(O) (A) or Ph(EtO)$_2$P(O) (B) | Ph$_3$P(O) | |
| 1 | Ph (**a**) | 1.15 [b] | 60 | EtOH | 100 [c] | 20 | 19 [d] | – | 48 | – |
| 2 | Ph (**a**) | 1.3 [e] | 60 | EtOH | 100 | 20 | 20 [d] | 2 (**A**) | 58 | – |
| 3 | EtO (**b**) | 1.15 [b] | 30 | EtOH | 96 | 30 | – | 66 (**B**) | n. r. | – |
| 4 | EtO (**b**) | 1.15 [b] | 30 | – | 81 | 69 | – | 12 (**B**) | n. r. | 41 (**6b**) |
| 5 | EtO (**b**) | 1.3 [e] | 45 | – | 100 | 74 | – | 16 (**B**) | n. r. | 60 (**6b**) |

[a] Based on $^{31}$P NMR. [b] In the presence of 5 mol% Pd(OAc)$_2$ catalyst. [c] 13% Ph$_2$P(O)OH, $\delta_P$ (CDCl$_3$) 28.2; [M + H]$^+$ = 219.0568, C$_{12}$H$_{12}$O$_2$P requires 219.0575. [d] $\delta_P$ (CDCl$_3$) 31.4, [M + H]$^+$ = 479.1329, C$_{30}$H$_{25}$O$_2$P$_2$ requires 479.1330. [e] In the presence of 10 mol% Pd(OAc)$_2$ catalyst. n. r.: not relevant.

The reaction with diethyl phosphite using 5 mol% of Pd(OAc)$_2$, 1.15 equivalents of the P-reagent and triethylamine in ethanol at 150 °C resulted in a mixture comprising 30% of 2-bromo-diethylphosphonoylbenzene (**6b**) along with 66% of diethyl phenylphosphonate (Table 7, entry 3). However, in a solvent-free manner, the side-reaction could be suppressed, although the conversion was not complete (Table 7, entry 4). The best run occurred, when 10 mol% of the catalyst precursor was used along with 1.3 equivalents of (EtO)$_2$P(O)H in the absence of ethanol for a longer irradiation time of 45 min (Table 7, entry 5). The bromophenylphosphonate (**6b**) could be isolated in 60% yield after chromatography. It can be noted that although 1,2-dibromobenzene is a sterically hindered substrate, its monophosphonoylation is possible.

The similar reactions of bromo-2-iodobenzene with Ph$_2$P(O)H and (EtO)$_2$P(O)H reagents were more clear-cut. The coupling with Ph$_2$P(O)H at 150 °C in ethanol took place so that the desired product **6a** was present in 65/66% portions in the mixture, and was prepared in a yield of 45%. The remaining part comprised ~10% of ethyl diphenylphosphinate, ~11% of Ph$_3$P(O) and 15/12% of Ph$_2$P(O)OH (Table 8, entries 1 and 2). Ph$_3$P(O) may have formed from **6a** by debromination. In another experiment, we applied 2.15 equivalents of Ph$_2$P(O) in order to synthesize bis-P(O)Ph$_2$ compound **7a**. After a reaction time of 1 h, the mixture comprised ~50% of the target product **7a**, along with ~10% of (EtO)Ph$_2$P(O) and 40% of Ph$_3$P(O). Due to the side-reactions, this approach is not efficient.

**Table 8.** P–C coupling reactions of 1-bromo-2-iodobenzene.

| Entry | Y | P-Reagent (equiv.) | t (min) | Solvent | Conversion (%) [a] | Product Composition (%) [a] | | | | Isolated Yield (%) |
|---|---|---|---|---|---|---|---|---|---|---|
| | | | | | | 6 | Ph$_2$(EtO)P(O) (A) or Ph(EtO)$_2$P(O) (B) | Ph$_3$P(O) | Ph$_2$P(O)OH | |
| 1 | Ph (**a**) | 1.15 [b] | 60 | EtOH | 100 | 65 | 9 (**A**) | 11 | 15 [c] | 45 (**6a**) |
| 2 | Ph (**a**) | 1.3 [d] | 60 | EtOH | 100 | 66 | 11 (**A**) | 11 | 12 [c] | – |
| 3 | EtO (**b**) | 1.3 [d] | 45 | EtOH | 95 | 36 | 36 (**B**) | n. r. | n. r. | – |
| 4 | EtO (**b**) | 1.3 [d] | 45 | – | 100 | 91 | 9 (**B**) | n. r. | n. r. | 75 (**6b**) |
| 5 | EtO (**b**) | 1.15 [b] | 30 | – | 85 | 69 | 16 (**B**) | n. r. | n. r. | 43 (**6b**) |

[a] Based on $^{31}$P NMR. [b] In the presence of 5 mol% Pd(OAc)$_2$ catalyst. [c] $\delta_P$ (CDCl$_3$) 28.2; [M + H]$^+$ = 219.0568, C$_{12}$H$_{12}$O$_2$P requires 219.0575. [d] In the presence of 10 mol% Pd(OAc)$_2$ catalyst. n. r.: not relevant.

It is noteworthy that the P–C coupling with (EtO)$_2$P(O)H was again more selective in the solvent-free variation. Performing the model reaction in ethanol, the proportion of arylphosphonate **6b** was only 36%, but in the absence of any solvent, the resulting mixture contained 91% of the target compound (**6b**) along with 9% of diethyl phenylphoshonate (Table 8, entries 3 and 4). Using a smaller amount of catalyst precursor and P-ligand, the conversion was not so efficient (Table 8, entry 5).

In order to prepare a mixed derivative, 4-bromophenyl-diphenylphosphine oxide **1a** was reacted with 1.15 equivalents of (EtO)$_2$P(O)H in the presence of 5 mol% of Pd(OAc)$_2$ and 1.1 equivalents of NEt$_3$ in EtOH at 150 °C under MW irradiation. The useful conversion of ca. 80% allowed a 51% isolated yield of 4-diethylphosphonoylphenyl-diphenylphosphine oxide **8** after chromatography (Scheme 1).

**Scheme 1.** P–C coupling of 4-bromophenyl-(diphenylphosphine oxide) with (EtO)$_2$P(O)H.

Finally, 3-bromophenyl-diphenylphosphine oxide (**3a**) was reacted with 1.15 equivalents of Ph$_2$P(O)H and (EtO)$_2$P(O)H under optimum conditions (5 mol% Pd(OAc)$_2$, 1.1 equivalents NEt$_3$ in EtOH at 150 °C), involving MW assistance. Although the formation of a few by-products was inevitable, the efficiency was quite good. Products **4a** and **9** were obtained in yields of 67% and 64%, respectively (Table 9, entries 1 and 2).

**Table 9.** P–C coupling reactions of 3-bromophenyl(diphenylphosphine oxide) with $Ph_2P(O)H$ and $(EtO)_2P(O)H$ reagents.

| Entry | Y | Product Composition (%) [a] | | | | Isolated Yield (%) |
| --- | --- | --- | --- | --- | --- | --- |
| | | 4a or 9 | $(EtO)Ph_2P(O)$ | $Ph_3P(O)$ | $Ph_2P(O)OH$ | |
| 1 | Ph (a) | 91 (4a) | 3 | 4 | 2 | 67 (4a) |
| 2 | EtO (b) | 87 (9) | n. r. | 13 | n. r. | 64 (9) |

[a] Based on $^{31}P$ NMR. n. r.: not relevant.

The mono >P(O)Ar products (**1a**, **1b**, **3a**, **5a**, **5b**, **6a** and **6b**), as well as the bis (>P(O))arene derivatives (**2b**, **4a**, **8**, and **9**) were fully characterized by $^{31}P$, $^{13}C$, $^{1}H$ NMR spectral data along HRMS. The pulse programs of one-dimensional ($^{1}H$, $^{13}C$ and DEPTQ) and two-dimensional ($^{1}H$,$^{1}H$-COSY, $^{1}H$,$^{13}C$-HSQC, $^{1}H$,$^{13}C$-HMBC and $^{1}H$,$^{1}H$-ROESY) measurements were utilized during the structure elucidation of compounds **4a**, **6a**, **8** and **9**. The starting points of signal assignment were easily identifiable units of molecules: the methyl and methylene groups of the ethoxy moiety, and the triplet para hydrogens of the $P(Ph)_2$ unit. The remainder of the molecular structures was elucidated using the same NMR methods mentioned above, e.g.: COSY cross-peak between ortho and para hydrogens of the $P(Ph)_2$ unit, ROESY correlation of methylene hydrogens to some hydrogens of the disubstituted benzene ring, and the quaternary atoms by their HMBC correlations to hydrogens of own aromatic rings.

In summary, 12 compounds were synthesized, of which 3 (**5a**, **8** and **9**) are new derivatives. Compound **3a** was mentioned in a patent, however, no spectral characterization was provided [34]. $^{31}P$, $^{13}C$ and $^{1}H$ NMR spectra can be found in the Supplementary Materials section.

## 3. Experimental

### 3.1. General Information

The reactions were carried out in a CEM® Discover Model SP (300 W) focused microwave reactor (Buckingham, UK) equipped with a stirrer and a pressure controller using 80–100 W irradiation under isothermal conditions. The reaction mixtures were irradiated in sealed borosilicate glass vessels (with a volume of 10 mL) available from the supplier of CEM®. The reaction temperature was monitored by an external IR sensor.

The $^{31}P$, $^{13}C$ and $^{1}H$ NMR spectra were taken in $CDCl_3$ solution on a Bruker Avance 300/Avance 500 spectrometer (Rheinstetten, Germany) operating at 121.5/202.4, 75.5/125.7 and 300 / 500 MHz, respectively. The $^{31}P$ chemical shifts are downfield relative to $H_3PO_4$, while the $^{13}C$ and $^{1}H$ chemical shifts are downfield relative to TMS. The couplings are given in Hz. The exact mass measurements were performed using an Agilent 6545 Q-TOF mass spectrometer (Santa Clara, CA, USA) in high resolution, positive electrospray mode. The chemicals were purchased from Sigma-Aldrich (Louis, MO, United States).

### 3.2. Procedures for the P–C Coupling of 1,4- or 1,3-Dibromobenzene and Diphenylphosphine Oxide or Diethyl Phosphite (Table 2, Entries 1 and 3; Table 4, Entries 1, 2 and 4)

To 0.022 mmol of the $Pd(OAc)_2$ catalyst (4.8 mg) in 1 mL of ethanol or without any solvent were added 0.43 mmol of dibromobenzene [1,4-dibromobenzene: 0.10 g or 1,3-dibromobenzene: 0.052 mL], 0.49 mmol (0.10 g) or 0.92 mmol (0.19 g) of diphenylphosphine

oxide or 0.49 mmol (0.063 mL) of diethyl phosphite, and 0.47 mmol (0.066 mL) or 0.95 mmol (0.13 mL) of triethylamine. Then, the mixture was irradiated in a closed vial in the MW reactor at 120 °C or 150 °C for the times shown in Table 2, entry 1 and Table 4, entries 1 and 2, or Table 2, entry 3 and Table 4, entry 4. The reaction mixture was diluted with 3 mL of EtOH, filtrated, and the residue obtained after evaporation was passed through a thin (2–3 cm) layer of silica gel using ethyl acetate as the eluent. The crude product was analyzed by $^{31}$P NMR spectroscopy, then it was purified by column chromatography (silica gel, and hexane–acetone 6:4 as the eluent) to afford compounds **1a**, **3a** and **4a** (Table 2, entry 1 and Table 4, entries 1 and 2), or phosphonates **1b** and **3b** (Table 2, entry 3 and Table 4, entry 4).

### 3.3. Procedures for the P–C Coupling of 4-Bromo- or 3-Bromo-Iodobenzenes and Diphenylphosphine Oxide or Diethyl Phosphite (Table 3, Entries 2, 4 and 5; Table 5, Entries 1, 3 and 4)

To 0.022 mmol of the Pd(OAc)$_2$ catalyst (4.8 mg) in 1 mL of ethanol or without any solvent were added 0.43 mmol of bromo-iodobenzenes [4-bromo-iodobenzene: 0.12 g, or 3-bromo-iodobenzene: 0.055 mL], 0.49 mmol (0.10 g) or 0.92 mmol (0.19 g) of diphenylphosphine oxide or 0.49 mmol (0.063 mL) or 0.99 mmol (0,13 mL) of diethyl phosphite, and 0.47 mmol (0.066 mL) or 0.95 mmol (0.13 mL) of triethylamine. Then, the mixture was irradiated in a closed vial in the MW reactor at 100 or 120 °C for 25–30 min (Table 3, entry 2 and Table 5, entries 1 and 3, or Table 3, entries 4 and 5, and Table 5, entry 4). The work-up and analysis along with the purification was done as described under 3.2 to furnish compounds **1a**, **3a** and **4a** (Table 3, entry 2 and Table 5, entries 1 and 3), or phosphonates **1b, 2b** and **3b** (Table 3, entries 4 and 5, and Table 5, entry 4).

### 3.4. Procedure for the P–C Coupling of 3-Chloro-Bromobenzene and Diphenylpshosphine Oxide or Diethyl Phosphite (Table 6, Entry 1 and 2)

To 0.022 mmol of the Pd(OAc)$_2$ catalyst (4.8 mg) in 1 mL of ethanol were added 0.43 mmol (0.051 mL) 3-chloro-bromobenzene, 0.49 mmol of >P(O)H-reagent [diphenylphosphine oxide: 0.10 g, or diethyl phosphite: 0.063 mL], and 0.47 mmol (0.066 mL) of triethylamine. Then, the mixture was irradiated in a closed vial in the MW reactor at 120 °C for 60 or 45 min. 3 mL of EtOH was added to the mixture, it was filtrated, and the residue obtained after evaporation was passed through a thin (2–3 cm) layer of silica gel using ethyl acetate as the eluent. The crude product was purified by column chromatography (silica gel, and dichloromethane–methanol 96:4 or ethyl acetate as the eluent) to give 0.11 g (80%) of product **5a** or 0.088 g (82%) of product **5b**.

### 3.5. The Preparation of Diethyl 2-Bromophenylphosphonate 6b (Table 8, Entry 4)

To 0.043 mmol of the Pd(OAc)$_2$ catalyst (9.6 mg) were added 0.43 mmol (0.055 mL) of 2-bromo-iodobenzene, 0.56 mmol (0.072 mL) of diethyl phosphite, and 0.47 mmol (0.066 mL) of triethylamine. Then, the mixture was irradiated in a closed vial in the MW reactor at 150 °C for 45 min. The crude reaction mixture was diluted with 3 mL of EtOH, filtrated, and the residue obtained after concentration was passed through a thin (2–3 cm) layer of silica gel using ethyl acetate as the eluent. The crude product was analyzed by $^{31}$P NMR spectroscopy, and then it was purified by column chromatography (silica gel, and ethyl acetate as the eluent) to give phosphonate **6b** in a yield of 75% (0.12 g).

### 3.6. The Synthesis of 4-Diethylphosphonoylphenyl-Diphenylphosphine Oxide 8 (Scheme 1)

To a MW glass vessel were added 0.021 mmol of the Pd(OAc)$_2$ catalyst (4.7 mg), 0.42 mmol of (4-bromophenyl)-diphenylphosphine oxide **1a** (0.15 g), 0.48 mmol (0.062 mL) of diethyl phosphite, 0.46 mmol (0.064 mL) of triethylamine and 1 mL of ethanol. The mixture was irradiated in a closed vial in the MW reactor at 150 °C for 30 min. The reaction mixture was diluted with 3 mL of EtOH, filtrated, and the residue obtained after concentration was passed through a thin (2–3 cm) layer of silica gel using ethyl acetate as the eluent. The crude product was analyzed by $^{31}$P NMR spectroscopy, and then it was purified

by column chromatography (silica gel, and dichloromethane–methanol 97:3 as the eluent) to give 4-diethylphosphonoylphenyl-diphenylphosphine oxide **8** in a yield of 51% (0.09 g).

### 3.7. The Procedure for the P–C Coupling of 3-Bromophenyl-Diphenylphosphine Oxide 3a with Diphenylphosphine Oxide and Diethyl Phosphite (Table 9, Entry 1 and 2)

To a MW glass vessel were added 0.021 mmol of the Pd(OAc)$_2$ catalyst (4.7 mg), 0.42 mmol of (3-bromophenyl)-diphenylphosphine oxide **3a** (0.15 g), 0.48 mmol of >P(O)H-reagent [0.097 g of diphenylphosphine oxide or 0.062 mL of diethyl phosphite], 0.46 mmol (0.064 mL) of triethylamine and 1 mL of ethanol. Then, the mixture was irradiated in a closed vial in the MW reactor at 150 °C for 30 min. The reaction mixture was diluted with 3 mL of EtOH, filtrated, and the residue obtained after concentration was passed through a thin (2–3 cm) layer of silica gel using ethyl acetate as the eluent. The crude product was analyzed by $^{31}$P NMR spectroscopy, and then it was purified by column chromatography (silica gel, and dichloromethane–methanol 97:3 as the eluent) to give 0.13 g (67%) of 1,3-phenylenebis(diphenylphosphine oxide) **4a** or 0.11 g (64%) of 3-diethylphosphonoylphenyl-diphenylphosphine oxide **9**.

### 3.8. Spectral Data for the Compounds Isolated

*(4-Bromophenyl)-diphenylphosphine Oxide (1a)* (Table 2, Entry 1 and Table 3, Entry 2). Appearance: white crystals, mp. 157–158 °C; $^{31}$P NMR (CDCl$_3$, 202.4 MHz) δ 28.5, δ$_P$ [29] (CDCl$_3$, 162 MHz) 25.2, δ$_P$ [35] (CDCl$_3$, 162 MHz) 28.7; $^{13}$C NMR (CDCl$_3$, 125.7 MHz) δ 133.6 (d, *J* = 10.6, C2)[a], 132.2 (d, *J* = 2.8, C4'), 132.1 (d, *J* = 104.9, C1'), 132.0 (d, *J* = 9.9, C2')[b], 131.84 (d, *J* = 12.5, C3)[a], 131.76 (d, *J* = 104.3, C1), 128.7 (d, *J* = 12.2, C3')[b], 127.2 (d, *J* = 3.4, C4), [a,b] may be reversed, δ$_C$ [29] (CDCl$_3$, 100 MHz) 133.5 (d, *J* = 10.5), 132.1 (d, *J* = 2.8), 131.9 (d, *J* = 104.4), 131.9 (d, *J* = 10.0), 131.7 (d, *J* = 12.4), 131.6 (d, *J* = 107.1), 128.6 (d, *J* = 12.1), 128.1 (d, *J* = 3.4), δ$_C$ [35] (CDCl$_3$, 100 MHz) 133.6 (d, *J* = 10.6), 132.1 (d, *J* = 2.7), 132.0 (d, *J* = 10.0), 131.9 (d, *J* = 104.6), 131.8 (d, *J* = 12.4), 131.3 (d, *J* = 45.6), 128.6 (d, *J* = 12.2), 127.2 (d, *J* = 3.3); $^1$H NMR (CDCl$_3$, 500 MHz) δ 7.66–7.59 (m, 6H), 7.57–7.51 (m, 4H), 7.48–7.45 (m, 4H), δ$_H$ [29] (CDCl$_3$, 600 MHz) 7.72–7.59 (m, 6H), 7.58–7.52 (m, 4H), 7.49–7.46 (m, 4H), δ$_H$ [35] (CDCl$_3$, 400 MHz) 7.68–7.65 (m, 3H), 7.63–7.62 (m, 2H), 7.60–7.58 (m, 2H), 7.56–7.51 (m, 4H), 7.49–7.47 (m, 3H); [M + H]$^+$ = 357.0045 C$_{18}$H$_{15}$OPBr requires 357.0044.

*Diethyl 4-Bromophenylphosphonate (1b)* (Table 2, Entry 3 and Table 3, Entry 4). Appearance: colorless oil; $^{31}$P NMR (CDCl$_3$, 202.4 MHz) δ 17.8, δ$_P$ [32] (CDCl$_3$, 121 MHz) 18.3, δ$_P$ [36] (CDCl$_3$, 121 MHz) δ 16.7; $^{13}$C NMR (CDCl$_3$, 125.7 MHz) δ 133.3 (d, *J* = 10.7, C2)[a], 131.8 (d, *J* = 15.5, C3)[a], 127.54 (d, *J* = 190.5, C1), 127.53 (d, *J* = 4.2, C4), 62.3, (d, *J* = 5.5, CH$_2$), 16.3 (d, *J* = 6.4, CH$_3$), [a] may be reversed, δ$_C$ [32] (CDCl$_3$, 75 MHz) 133.4, 133.3, 132.0, 131.8, 127.7 (*J* = 188.59), 127.6, 62.4 (*J* = 5.02), 16.3 (*J* = 6.46), δ$_C$ [36] (CDCl$_3$, 75 MHz) 133.3 (*J* = 10.1), 131.7 (*J* = 16.5), 129.8 (*J* = 185.0), 127.5 (*J* = 3.9), 61.8 (*J* = 5.7), 16.3 (*J* = 5.7); $^1$H NMR (CDCl$_3$, 500 MHz) δ 7.72–7.68 (m, 2H), 7.65–7.62 (m, 2H), 4.21–4.06 (m, 4H, CH$_2$), 1.34 (t, *J* = 7.0, 6H, CH$_3$), δ$_H$ [32] (CDCl$_3$, 300 MHz) 7.60–7.72 (m, 4H), 4.10–4.20 (m, 4H), 1.32 (t, *J* = 7.2, 6H), δ$_H$ [36] (CDCl$_3$, 300 MHz) 7.62 (d, *J* = 8.94, 2H), 7.20 (d, *J* = 8.94, 2H), 3.84–3.86 (m, 4H), 1.17 (t, *J* = 7.2, 6H); [M + H]$^+$ = 292.9939 C$_{10}$H$_{15}$O$_3$PBr requires 292.9942.

*Tetraethyl 1,4-Phenylenebisphosphonate (2b)* (Table 3, Entry 5). Appearance: colorless oil; $^{31}$P NMR (CDCl$_3$, 202.4 MHz) δ 16.8, δ$_P$ [37] (121 MHz, CDCl$_3$) 17.5; $^{13}$C NMR (CDCl$_3$, 125 MHz) δ 132.9 (dd, *J*$_1$ = 189.9, *J*$_2$ = 2.7, C1), 131.7–131.5 (m, C2), 62.5 (d, *J* = 5.4, CH$_2$), 16.3 (d, *J* = 6.4, CH$_3$), δ$_C$ [37] (75 MHz, CDCl$_3$) 134.1 (d, *J* = 3.1), 131.8–131.4 (m), 63.3–62.3 (m), 16.5–16.2 (m); $^1$H NMR (CDCl$_3$, 500 MHz) δ 7.95–7.91 (m, 4H), 4.24–4.09 (m, 8H, CH$_2$), 1.36 (t, *J* = 7.1, 12H, CH$_3$), δ$_H$ [37] (300 MHz, CDCl$_3$) 7.97–7.84 (m, 4H), 4.24–4.01 (m, 8H), 1.39–1.28 (m, 12H); [M + H]$^+$ = 351.1121 C$_{14}$H$_{25}$O$_6$P$_2$ requires 351.1126.

*3-Bromophenyl-diphenylphosphine Oxide (3a)* (Table 4, Entry 1 and Table 5, Entry 1). Appearance: white crystals, mp. 92–93 °C; $^{31}$P NMR (CDCl$_3$, 202.4 MHz) δ 28.0; $^{13}$C NMR (CDCl$_3$, 125.7 MHz) δ 135.4 (d, *J* = 100.8, C1), 135.0 (d, *J* = 2.5, C4), 134.6 (d, *J* = 10.5, C2), 132.3 (d, *J* = 2.8, C4'), 132.0 (d, *J* = 10.0, C2')[a], 131.8 (d, *J* = 105.0, C1'), 130.5 (d, *J* = 9.5, C6), 130.2 (d, *J* = 12.6, C5), 128.7 (d, *J* = 12.3, C3')[a], 123.2 (d, *J* = 15.2, C3), [a] may be reversed;

[1]H NMR (CDCl$_3$, 500 MHz) δ 7.84–7.81 (m, 1H), 7.69–7.64 (m, 5H), 7.60–7.56 (m, 3H), 7.51–7.47 (m, 4H), 7.36–7.32 (m, 1H); [M + H]$^+$ = 357.0046 C$_{18}$H$_{15}$OPBr requires 357.0044.

*1,3-Phenylenebis(diphenylphosphine Oxide) (4a)* (Table 5, Entry 3). Appearance: colorless oil; [31]P NMR (CDCl$_3$, 202.4 MHz) δ 28.5, δ$_P$ [27] (CDCl$_3$,162 MHz) 30.5; [13]C NMR (CDCl$_3$, 125.7 MHz) δ 135.5 (dd, $J_1$ = 10.1, $J_2$ = 3.1, C3), 135.4 (t, $J$ = 11.2, C1), 133.6 (dd, $J_1$ = 101.8, $J_2$ = 10.7, C2), 132.2 (d, $J$ = 2.3, C4′)$^a$, 131.95 (d, $J$ = 10.2, C2′)$^a$, 131.7 (d, $J$ = 105.1, C1′), 128.95 (t, $J$ = 11.3, C4), 128.6 (d, $J$ = 12.5, C3′)$^a$, $^a$ may be reversed, δ$_C$ [27] (CDCl$_3$, 100 MHz) 135.2–135.4 (m, 2C), 135.1, 133.5 (dd, $J_1$ = 101.7, $J_2$ = 10.7), 132.0, 131.8 (d, $J$ = 10.3), 131.5 (d, $J$ = 104.9), 128.8 (t, $J$ = 11.2), 128.4 (d, $J$ = 12.6), 127.1; [1]H NMR (CDCl$_3$, 500 MHz) δ 7.96 (ddm, $J_1$ = 12.5, $J_2$ = 7.7, 2H), 7.69 (tt, $J_1$ = 11.7, $J_2$ = 1.5, 1H), 7.62 (tt, $J_1$ = 7.7, $J_2$ = 2.5, 1H), 7.58 (dd, $J_1$ = 12.1, $J_2$ = 7.9, 8H), 7.53 (t, $J$ = 7.4, 4H), 7.41 (td, $J_1$ = 7.7, $J_2$ = 2.8, 8H), δ$_H$ [27] (CDCl$_3$, 400 MHz) 7.93–7.98 (m, 2H), 7.71 (t, $J$ = 11.7, 1H), 7.50–7.63 (m, 13H), 7.38–7.43 (m, 8H); [M + H]$^+$ = 479.1327 C$_{30}$H$_{25}$O$_2$P$_2$ requires 479.1330.

*Diethyl 3-Bromophenylphosphonate (3b)* (Table 4, Entry 4 and Table 5, Entry 4). Appearance: colorless oil; [31]P NMR (CDCl$_3$, 202.4 MHz) δ 16.2; [13]C NMR (CDCl$_3$, 125.7 MHz) δ 135.6 (d, $J$ = 2.8, C4), 134.7 (d, $J$ = 10.7, C2)$^a$, 131.2 (d, $J$ = 187.3, C1), 130.4 (d, $J$ = 6.0, C6), 130.3 (d, $J$ = 12.8, C5)$^a$, 123.0 (d, $J$ = 19.7, C3), 62.6 (d, $J$ = 5.5, CH$_2$), 16.5 (d, $J$ = 6.4, CH$_3$), $^a$ may be reversed, δ$_C$ [33] (101 MHz, CDCl$_3$) 135.4 (d, $J$ = 3.0), 134.5 (d, $J$ = 10.6), 131.1 (d, $J$ = 186), 130.2 (d, $J$ = 4.9), 130.1 (d, $J$ = 11.7), 122.9 (d, $J$ = 19.8), 62.4 (d, $J$ = 5.5), 16.3 (d, $J$ = 6.4); [1]H NMR (CDCl$_3$, 500 MHz) δ 7.98–7.95 (m, 1H), 7.79–7.74 (m, 1H), 7.71–7.69 (m, 1H), 7.39–7.35 (m, 1H), 4.23–4.08 (m, 4H, CH$_2$), 1.36 (t, $J$ = 7.1, 6H, CH$_3$), δ$_H$ [33] (400 MHz, CDCl$_3$) 7.93 (d, $J$ = 13.6, 1H), 7.72 (dd, $J_1$ = 12.9, $J_2$ = 7.6, 1H), 7.66 (d, $J$ = 8.0, 1H), 7.33 (td, $J_1$ = 7.8, $J_2$ = 4.8, 1H), 4.21–4.00 (m, 4H), 1.32 (t, $J$ = 7.1, 6H); [M + H]$^+$ = 292.9937 C$_{10}$H$_{15}$O$_3$PBr requires 292.9942.

*Diphenyl-3-chlorophenylphosphine Oxide (5a)* (Table 6, Entry 1). Appearance: white crystals, mp. 75–76 °C; [31]P NMR (CDCl$_3$, 202.4 MHz) δ 28.1; [13]C NMR (CDCl$_3$, 125.7 MHz) δ 135.1 (d, $J$ = 101.3, C1), 135.0 (d, $J$ = 15.6, C3), 132.3 (d, $J$ = 2.8, C4′), 132.1 (d, $J$ = 2.7, C4), 132.0 (d, $J$ = 10.0, C2′)$^a$, 131.84 (d, $J$ = 10.7, C2)$^b$, 131.82 (d, $J$ = 105.1, C1′), 130.1 (d, $J$ = 9.5, C6)$^b$, 129.9 (d, $J$ = 12.9, C5), 128.7 (d, $J$ = 12.3, C3′)$^a$, $^{a,b}$ may be reversed; [1]H NMR (CDCl$_3$, 500 MHz) δ 7.68–7.64 (m, 5H), 7.59–7.55 (m, 3H), 7.53–7.47 (m, 5H), 7.43–7.39 (m, 1H); [M + H]$^+$ = 313.0547 C$_{18}$H$_{15}$OPCl requires 313.0549.

*Diethyl 3-Chlorophenylphosphonate (5b)* (Table 6, Entry 2). Appearance: colorless oil; [31]P NMR (CDCl$_3$, 202.4 MHz) δ 16.5, δ$_P$ [17] (CDCl$_3$) 17.5; [13]C NMR (CDCl$_3$, 125.7 MHz) δ 134.8 (d, $J$ = 20.3, C3), 132.5 (d, $J$ = 3.0, C4), 131.6 (d, $J$ = 10.7, C2)$^a$, 130.8 (d, $J$ = 187.8, C1), 129.9 (d, $J$ = 16.5, C5), 129.8 (d, $J$ = 9.2, C6)$^a$, 62.4 (d, $J$ = 5.5, CH$_2$), 16.3 (d, $J$ = 6.4, CH$_3$), $^a$ may be reversed, δ$_C$ [17] (CDCl$_3$) 134.7 (d, $J$ = 20.3, C2), 132.4 (d, $J$ = 3.0, C4), 131.6 (d, $J$ = 10.7, C3), 130.7 (d, $J$ = 187.9, C1), 129.74 (d, $J$ = 16.3, C6), 129.69 (d, $J$ = 9.2, C5), 62.3 (d, $J$ = 5.5, CH$_2$), 16.2 (d, $J$ = 6.4, CH$_3$); [1]H NMR (CDCl$_3$, 500 MHz) δ 7.83–7.68 (m, 2H), 7.56–7.53 (m, 1H), 7.46–7.39 (m, 1H), 4.25–4.05 (m, 4H, CH$_2$), 1.36 (t, $J$ = 7.1, 6H, CH$_3$), δ$_H$ [17] (CDCl$_3$) 7.81–7.59 (m, 2H, ArH), 7.51–7.43 (m, 1H, ArH), 7.40–7.31 (m, 1H, ArH), 4.20–3.96 (m, 4H, OCH$_2$), 1.30 (t, $J$ = 7.1, 6H, CH$_3$); [M + H]$^+$ = 249.0448 C$_{10}$H$_{15}$O$_3$PCl requires 249.0447.

*(2-Bromophenyl)-diphenylphosphine Oxide (6a)* (Table 8, Entry 1). Appearance: white crystals; [31]P NMR (CDCl$_3$, 202.4 MHz) δ 30.5, δ$_P$ [25] (CDCl$_3$, 200 MHz) 30.6, δ$_P$ [24] (CDCl$_3$, 162 MHz) 32.2; [13]C NMR (CDCl$_3$, 125.7 MHz) δ 136.0 (d, $J$ = 10.5, C3)$^a$, 134.9, (d, $J$ = 7.5, C6)$^a$, 133.4 (d, $J$ = 2.4, C4), 133.0 (d, $J$ = 104.6, C1), 132.1 (d, $J$ = 10.0, C2′)$^b$, 132.0 (d, $J$ = 2.8, C4′), 131.7 (d, $J$ = 108.0, C1′), 128.6 (d, $J$ = 12.5, C3′)$^b$, 127.0 (d, $J$ = 11.1, C5)$^a$ 126.9 (d, $J$ = 4.7, C2), $^{a,b}$ may be reversed, δ$_C$ [25] (CDCl$_3$, 75 MHz) 136.1 (d, $J$ = 10.4), 135.0 (d, $J$ = 7.7), 133.6 (d, $J$ = 2.2), 133.2 (d, $J$ = 104.3), 132.3 (d, $J$ = 9.9), 132.1 (d, $J$ = 2.7), 131.9 (d, $J$ = 107.6), 128.7 (d, $J$ = 12.6), 127.1 (d, $J$ = 11.5), 127.1 (d, $J$ = 4.9), δ$_C$ [24] (CDCl$_3$, 100 MHz) 136.3 (d, $J$ = 10.4), 135.2 (d, $J$ = 8.0), 133.7 (d, $J$ = 2.4), 133.5 (d, $J$ = 104.7), 132.5 (d, $J$ = 9.6), 132.3 (d, $J$ = 2.4), 132.2 (d, $J$ = 107.9), 128.9 (d, $J$ = 12.0), 127.3 (d, $J$ = 11.2), 127.3; [1]H NMR (CDCl$_3$, 500 MHz) δ 7.74–7.71 (m, 4H), 7.70–7.67 (m, 1H), 7.59–7.55 (m, 2H), 7.50–7.46 (m, 4H), 7.41–7.32 (m, 3H,) δ$_H$ [25] (CDCl$_3$, 500 MHz) 7.75–7.65 (m, 5H), 7.58–7.53 (m, 2H), 7.60–7.45

(m, 4H), 7.41–7.30 (m, 3H), $\delta_H$ [24] (CDCl$_3$, 400 MHz) 7.75–7.65 (m, 5H), 7.58–7.31 (m, 9H); [M + H]$^+$ = 357.0045 C$_{18}$H$_{15}$OPBr requires 357.0044.

*Diethyl 2-Bromophenylphosphonate (**6b**)* (Table 8, Entry 4). Appearance: colorless oil; $^{31}$P NMR (CDCl$_3$, 202.4 MHz) δ 14.8, $\delta_P$ [26] (CDCl$_3$, 121 MHz) 15.4; $^{13}$C NMR (CDCl$_3$, 125.7 MHz) δ 136.3 (d, *J* = 8.3, C6)$^a$, 134.3 (d, *J* = 11.2, C3)$^a$, 133.6 (d, *J* = 2.7, C4), 129.5 (d, *J* = 192.0, C1), 126.9 (d, *J* = 13.6, C5)$^a$, 125.2 (d, *J* = 3.8, C2), 62.6 (d, *J* = 5.6, CH$_2$), 16.3 (d, *J* = 6.5, CH$_3$), $^a$ may be reversed, $\delta_C$ [26] (CDCl$_3$, 75 MHz) 136.2 (d, *J* = 8.3, Ar-CH), 133.5 (d, *J* = 2.7, Ar-CH), 129.3 (d, *J* = 192.0, Ar-qC), 126.8 (d, *J* = 13.6, Ar-CH), 125.1 (d, *J* = 3.9, Ar-qC), 62.5 (d, *J* = 5.6 OCH$_2$CH$_3$), 16.2 (d, *J* = 6.6 OCH$_2$CH$_3$); $^1$H NMR (CDCl$_3$, 500 MHz) δ 8.07–8.02 (m, 1H), 7.71–7.68 (m, 1H), 7.45–7.38 (m, 2H), 4.27–4.12 (m, 4H, CH$_2$), 1.39 (t, *J* = 7.1, 6H, CH$_3$), $\delta_H$ [26] (CDCl$_3$, 300 MHz) 8.05–7.96 (m, 1H, ArH), 7.70–7.62 (m, 1H, ArH), 7.43–7.33 (m, 2H, ArH), 4.28–4.04 (m, 4H, OCH$_2$CH$_3$), 1.35 (td, $J_1$ = 7.1, $J_2$ = 0.5) 6H, OCH$_2$CH$_3$); [M + Na]$^+$ = 314.9761 C$_{10}$H$_{14}$O$_3$PBrNa requires 314.9762.

*4-Diethylphosphonoylphenyl-diphenylphosphine Oxide (**8**)* (Scheme 1). Appearance: colorless oil; $^{31}$P NMR (CDCl$_3$, 202.4 MHz) δ 28.4 (m, P(C$_6$H$_5$)$_2$), 16.8 (d; *J* = 3.7, P(OCH$_2$CH$_3$)$_2$); $^{13}$C NMR (CDCl$_3$, 125.7 MHz) δ 137.3 (dd, $J_1$ = 100.5, $J_2$ = 3.0, C1), 132.6 (dd, $J_1$ = 186.9, $J_2$ = 2.9, C4), 132.3 (d, *J* = 2.8, C4′), 132.1 (d, *J* = 10.0, C2′)$^a$, 132.0 (dd, *J* = 14.9, *J* = 9.8, C2), 131.7 (d, *J* ~ 105, C1′), 131.6 (dd, $J_1$ = 11.8, $J_2$ = 10.0, C3), 128.7 (d, *J* = 12.2, C3′)$^a$, 62.5 (d, *J* = 5.6, CH$_2$), 16.4 (d, *J* = 6.4, CH$_3$), $^a$ may be reversed; $^1$H NMR (CDCl$_3$, 500 MHz) 7.90 (ddd, $J_1$ = 12.8, $J_2$ = 8.3, $J_3$ = 2.5, 2H), 7.78 (ddd, $J_1$ = 11.6, $J_2$ = 8.1, $J_3$ = 3.9, 2H), 7.67 (ddd, $J_1$ = 12.1, $J_2$ = 7.8, $J_3$ = 1,3, 4H), 7.58 (tm, *J* = 7.5, 2H), 7.49 (td, $J_1$ = 7.6, $J_2$ = 2.9, 4H), 4.18 (m) 4.10 (m, 4H, CH$_2$), 1.34 (t, *J* = 7.1, 6H, CH$_3$); [M + H]$^+$ = 415.1228 C$_{22}$H$_{25}$O$_4$P$_2$ requires 415.1228.

*3-Diethylphosphonoylphenyl-diphenylphosphine Oxide (**9**)* (Table 9, Entry 2). Appearance: colorless oil; $^{31}$P NMR (CDCl$_3$, 202.4 MHz) δ 28.3 (m, P(C$_6$H$_5$)$_2$), 16.7 (d, *J* = 5.0, P(OCH$_2$CH$_3$)$_2$); $^{13}$C NMR (CDCl$_3$, 125.7 MHz) δ 135.8 (dd, $J_1$ = 9.7, $J_2$ = 2.8, C6), 135.2 (dd, $J_1$ = 10.0, $J_2$ = 2.6, C4), 135.0 (t, *J* = 10.6, C2), 133.6 (dd, $J_1$ = 102.1, $J_2$ = 13.7, C1), 132.3 (d, *J* = 2.9, C4′), 132.1 (d, *J* = 10.0, C2′)$^a$, 132.0 (d, *J* = 104.8, C1′), 129.5 (dd, $J_1$ = 188.6, $J_2$ = 11.3, C3), 128.75 (dd, $J_1$ = 14.2, $J_2$ = 11.2, C), 128.7 (d, *J* = 12.3, C3′)$^a$, 62.5 (d, *J* = 5.7, CH$_2$), 16.3 (d, *J* = 6.4, CH$_3$), $^a$ may be reversed; $^1$H NMR (CDCl$_3$, 500 MHz) 8.06 (t, *J* = 12.6, 1H), 8.01 (ddq, $J_1$ = 13.1, $J_2$ = 7.7, $J_3$ = 1.5, 1H), 7.89 (ddq, $J_1$ = 11.7, $J_2$ = 7.8, $J_3$ = 1.5, 1H), 7.67 (ddd, $J_1$ = 12.1, $J_2$ = 8.0, $J_3$ = 1.4, 4H), 7.59 (tt, $J_1$$^b$, $J_2$ = 3.3, 1H), 7.57 (tq, $J_1$ = 7.5, $J_2$ = 1.5, 2H), 7.48 (td, $J_1$ = 7.6, $J_2$ = 2.9, 4H), 4.12 (m) 4.05 (m, 4H, CH$_2$), 1.27 (t, *J* = 7.1, 6H, CH$_3$), $^b$ the coupling could not be detected due to overlapping signals; [M + Na]$^+$ = 437.1042 C$_{22}$H$_{24}$O$_4$P$_2$Na requires 437.1048.

## 4. Conclusions

A Hirao P–C cross coupling protocol using the excess of P-reagents as the P-ligands under MW irradiation was applied to the synthesis of bromophenylphosphine oxides and phosphonates in order to provide valuable starting materials for further transformations. A collection of 1,4-, 1,3- and 1,2-dibromobenzenes and the corresponding bromo-iodo derivatives were reacted with diphenylphosphine oxide and diethyl phosphite using Pd(OAc)$_2$ as the catalyst precursor and an excess of the Y$_2$P(O)H reagent (Y = Ph, EtO) as the P-ligand via its Y$_2$POH tautomeric form. The bromophenylphosphine oxides and phosphonates, in most cases, could be obtained in reasonable yields, however, the reaction of the ortho-dihalogenobenzenes was reluctant. The bromo-iodobenzenes could be replaced by the cheaper dibromo derivatives. In a few cases, the bis(>P(O)-benzene) species were also prepared, either directly, or from the isolated mono(>P(O)-bromophenyl) derivative. In all, 12 products, of which 3 are new, were prepared and characterized.

**Supplementary Materials:** The supporting information including the $^{31}$P, $^1$H, and $^{13}$C NMR spectra of the products can be downloaded at: https://www.mdpi.com/article/10.3390/catal12101080/s1.

**Author Contributions:** Conceptualization, G.K.; methodology, B.H.; formal analysis, P.R.V., A.S. and L.D.; investigation, B.H. and N.Á.S.; data curation, A.S. and L.D.; writing—original draft preparation, G.K. and B.H.; writing—review and editing, G.K.; supervision, G.K.; project administration, G.K.; funding acquisition, G.K. All authors have read and agreed to the published version of the manuscript.

**Funding:** This project was supported by the National Research, Development and Innovation Office (K134318).

**Data Availability Statement:** Not relevant.

**Conflicts of Interest:** The authors declare no conflict of interest.

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
