# Peer review of "Pd-Catalyzed Hirao P–C Coupling Reactions with Dihalogenobenzenes without the Usual P-Ligands under MW Conditions"

_catalysts, doi:10.3390/catal12101080_

Round 1
Reviewer 1 Report
In this manuscript, the authors described a microwave assisted method to prepare diethyl phosphonates and phosphine oxides derivatives from dihalobenzenes using Pd(0Ac)2 as catalyst and Et3N as base. In the introduction, they compared the previous reports in the area, showing that most of the methods require P-containing ligands.
Using the optimized conditions, with or without EtOH as solvent, they were able to prepare 12 derivatives with moderate to good isolated yields (up to 82%). The synthesized compounds were characterized by NMR and HRMS. The authors cited the characterization of the compounds very briefly only at the end of the results and discussion section (page 10). In the conclusions (page 15), the authors stated that 5 new products were prepared, however only 3 are really novel, since compounds 3a and 5b were already described in literature. The references should be cited in the experimental.
The experimental should contain only one (or two maximum) general optimized procedure for preparation of the two classes of compounds (phosphonates and phosphine oxides).
The manuscript writing should be improved as highlighted in yellow. For example, in several phrases, the authors used the expression “somewhat”, which is not appropriate. In tables 2-9, they used “n. r.” but the meaning was not specified. Some paragraphs are confusing and should be rewritten as on page 8.
In the support information, the authors should include legends and expansions of the NMR spectra in order to facilitated their analysis and comparisons with the previously reported ones.

Author Response
Replies to Referee 1
"The synthesized compounds were characterized by NMR and HRMS. The authors cited the characterization of the compounds very briefly only at the end of the results and discussion section (page 10)."
The characterization of the compounds prepared was indeed brief at the end of the R & D part. However, in the Experimental part there are responsible assignments occasionally confirmed by two dimensional measurements.
"In the conclusions (page 15), the authors stated that 5 new products were prepared, however only 3 are really novel, since compounds 3a and 5b were already described in literature. The references should be cited in the experimental."
O yes, we have overlooked that species 5b was described by us earlier in ref [17]. At the same time, phosphine oxide 3a was just mentioned in a patent (new ref [35]), but no spectral data were provided. Although this would caunt as a new derivative, for your sake, we classified it as a known compound. See the end of the R & D and the Conclusions part.
"The experimental should contain only one (or two maximum) general optimized procedure for preparation of the two classes of compounds (phosphonates and phosphine oxides)."
The 4 general procedures were comprised into 2 ones. However, the special (individual) methods had to be kept.
"The manuscript writing should be improved as highlighted in yellow. For example, in several phrases, the authors used the expression “somewhat”, which is not appropriate. In tables 2-9, they used “n. r.” but the meaning was not specified. Some paragraphs are confusing and should be rewritten as on page 8."
Thanks for spotting the bothering mistakes they were corrected. Even the “somewhat”-s were eliminated, although this word might be tolerable. “n. r.” means “not relevant”. We tried to do our best to be understandable.
"In the support information, the authors should include legends and expansions of the NMR spectra in order to facilitated their analysis and comparisons with the previously reported ones."
It is not usual to include expanded versions of routine NMR spectra in the Supplementary Information. These are sent only on the request of the readers. On your request a pdf file comprising typical expanded spectra have been attached.
Finally let us thank you for the bettering remarks.

Reviewer 2 Report
The manuscript describes a microwave assisted palladium catalyzed C-P coupling, and the P-containing compounds serve as both ligand and substrate, with some selectivity between the -Br and -I. This work provides an improved reaction condition for the efficient C-P coupling that of great interests in the field of P-containing ligand synthesis. However, the presentation of this manuscript is not scholarly, for example, the abstract should discuss the contents of work that the author has done, along with the novelty, significance of the work, rather than discuss the “literature survey”. Likewise, in page 2, the author introduces a lot of (1 page) previous reaction condition which is inappropriate and distract the main idea of the paper. Besides, there are some arguments that I would like the authors to consider.
1. Page 2, line 52, questioning the issues of facticity of the precedent in public is an extremely serious decision. If the reaction cannot be reproduced, please double check if all of the chemicals are quality, the amount of residual water in solvent is controlled, the reaction procedure and details are correct, etc. And also try to inquire the issues to the corresponding author of that paper rather than directly make the conclusion in your manuscript.
2. The Table 1 should be placed to Page 2 for the convenience of reading;
3. There is no need to place the reference number in Table 1, and the types of the yield should be indicated (isolated yield/NMR yield/GC/HPLC, etc).
4. The describe “other” in Table 1 is not scholarly, “additional conditions” should be more suitable here.
5. Page 2, as I mentioned above, do not distract the main idea of your manuscript by introducing too much details of the previous publications. If there is necessary to discuss, then using a two-paragraph-length brief summarization should be sufficient. Moreover, the author should try to strengthen the significance of this manuscript by adding a brief introduction of the advantage of microwave assisted C-P coupling reactions.
6. All of the catalyst loading, such as in line 91 and table 2-9, should be 5 mol % rather than 5 %.
7. All of the reaction scale and type of yield should be indicated in Table 2-9.
8. line 207, 150 C should be 150 oC.
9. In the section of experimental, the 0.022 mmol of Pd(OAc)2 should be 0.0048 g rather than 0.048 g, and I consider the unit "mg & ul " should be more suitable than “g & ml” here.
10. The author should at least conduct a gram-scale experiment to ensure the utility of this methodology.
11. The selectivity of the reaction between two halogen is one of the most important topics in this work, however, in the supplementary information, there is no any evidence for this part. The author should provide at least one evidence (eg. NMR/HPLC/GC) to indicate this selectivity.
In summary, this reviewer consider the results are useful, but I would like to see a more scholarly presentation before giving a final judgement on this.
Author Response
Replies to Referee 2
"... However, the presentation of this manuscript is not scholarly, for example, the abstract should discuss the contents of work that the author has done, along with the novelty, significance of the work, rather than discuss the “literature survey”."
The Abstract was well extended with the novelty and the significance.
"Likewise, in page 2, the author introduces a lot of (1 page) previous reaction condition which is inappropriate and distract the main idea of the paper. Later on: 5a. Page 2, as I mentioned above, do not distract the main idea of your manuscript by introducing too much details of the previous publications. If there is necessary to discuss, then using a two-paragraph-length brief summarization should be sufficient."
Our understanding is that the precedents of the topic should be presented in a depth that is justified by the earlier results. If we omitted this literature survey part, the reader would not be able to compare our results with the precedents. So we ask you to allow to keep literature summary written with an utmost care.
"Besides, there are some arguments that I would like the authors to consider.
- Page 2, line 52, questioning the issues of facticity of the precedent in public is an extremely serious decision. If the reaction cannot be reproduced, please double check if all of the chemicals are quality, the amount of residual water in solvent is controlled, the reaction procedure and details are correct, etc. And also try to inquire the issues to the corresponding author of that paper rather than directly make the conclusion in your manuscript."
We were more careful in the revised version to comment on our unsuccessful attempt to reproduce the coupling with Pd(OAc)2/PPh3 [27]. Please know that we applied the same conditions, and performed 3 independent experiments. There is a significant steric hindrance due to the adjacent bromine atom.
- "The Table 1 should be placed to Page 2 for the convenience of reading;"
The final arrangement of the Schemes/Tables will be done by the Editorial staff. The accepted version always undergoes significant changes regarding the lay-out.
"3a. There is no need to place the reference number in Table 1"
Why not to supply the literature data in the table by the appropriate references?
""3b. the types of the yield should be indicated (isolated yield/NMR yield/GC/HPLC, etc). Later on: 7. All of the reaction scale and type of yield should be indicated in Table 2-9."
We always ment “isolated yield”. This was inserted in the Tables. Our view is that “NMR yield, GC/HPLC etc.” may not make much sense.
- "The describe “other” in Table 1 is not scholarly, “additional conditions” should be more suitable here."
“Additional Conditions” was written instead of non-scholar “Other”.
"5b. Moreover, the author should try to strengthen the significance of this manuscript by adding a brief introduction of the advantage of microwave assisted C-P coupling reactions."
This was done in the Introduction.
- "All of the catalyst loading, such as in line 91 and table 2-9, should be 5 mol % rather than 5 %."
“5%” was corrected to “5 mol%” throughout.
- "line 207, 150 C should be 150 oC."
“150 C” was corrected to “150 °C”.
- "In the section of experimental, the 0.022 mmol of Pd(OAc)2 should be 0.0048 g rather than 0.048 g, and I consider the unit "mg & ul " should be more suitable than “g & ml” here."
Yes, “0.0048 g” is the correct quantity; now we gave it as “4.8 mg”.
- "The author should at least conduct a gram-scale experiment to ensure the utility of this methodology."
The size of the vial of the MW reactor limits the quantities. On your request, we performed experiment marked by Table 4/entry 1 on a 3-fold scale. The yield became somewhat better (76%). See page 5, line 142.
- "The selectivity of the reaction between two halogen is one of the most important topics in this work, however, in the supplementary information, there is no any evidence for this part. The author should provide at least one evidence (eg. NMR/HPLC/GC) to indicate this selectivity."
The selectivity of the mono- and biscouplings has been demonstrated by representative 31P NMR spectra along with LC-MS. See the attached pdf file!
Finally, we wish to you for your bettering comments and good-will.

Round 2
Reviewer 2 Report
The revised manuscript is improved for both science and scholarly presentation, this reviewer supports the acceptance of this manuscript in its current version.